# Lifted Inference Rules with Constraints

**Happy Mittal, Anuj Mahajan**
Dept. of Comp. Sci. & Engg.
I.I.T. Delhi, Hauz Khas
New Delhi, 110016, India
happy.mittal@cse.iitd.ac.in,
anujmahajan.iitd@gmail.com

**Vibhav Gogate**
Dept. of Comp. Sci.
Univ. of Texas Dallas
Richardson, TX 75080, USA
vgogate@hlt.utdallas.edu

**Parag Singla**
Dept. of Comp. Sci. & Engg.
I.I.T. Delhi, Hauz Khas
New Delhi, 110016, India
parags@cse.iitd.ac.in

## Abstract

Lifted inference rules exploit symmetries for fast reasoning in statistical relational models. Computational complexity of these rules is highly dependent on the choice of the constraint language they operate on and therefore coming up with the right kind of representation is critical to the success of lifted inference. In this paper, we propose a new constraint language, called *setineq*, which allows subset, equality and inequality constraints, to represent substitutions over the variables in the theory. Our constraint formulation is strictly more expressive than existing representations, yet easy to operate on. We reformulate the three main lifting rules: decomposer, generalized binomial and the recently proposed single occurrence for MAP inference, to work with our constraint representation. Experiments on benchmark MLNs for exact and sampling based inference demonstrate the effectiveness of our approach over several other existing techniques.

## 1 Introduction

Statistical relational models such as Markov logic [5] have the power to represent the rich relational structure as well as the underlying uncertainty, both of which are the characteristics of several real world application domains. Inference in these models can be carried out using existing probabilistic inference techniques over the propositionalized theory (e.g., Belief propagation, MCMC sampling, etc.). This approach can be sub-optimal since it ignores the rich underlying structure in the relational representation, and as a result does not scale to even moderately sized domains in practice.

Lifted inference ameliorates the aforementioned problems by identifying indistinguishable atoms, grouping them together and inferring directly over the groups instead of individual atoms. Starting with the work of Poole [21], a number of lifted inference algorithms have been proposed. These include lifted exact inference techniques such as lifted Variable Elimination (VE) [3, 17], lifted approximate inference algorithms based on message passing such as belief propagation [23, 14, 24], lifted sampling based algorithms [26, 12], lifted search [11], lifted variational inference [2, 20] and lifted knowledge compilation [10, 6, 9]. There also has been some recent work which examines the complexity of lifted inference independent of the specific algorithm used [13, 2, 8].

Just as probabilistic inference algorithms use various rules such as sum-out, conditioning and decomposition to exploit the problem structure, lifted inference algorithms use lifted inference rules to exploit the symmetries. All of them work with an underlying constraint representation that specifies the allowed set of substitutions over variables appearing in the theory. Examples of various constraint representations include weighted parfactors with constraints [3], normal form parfactors [17], hypercube based representations [24], tree based constraints [25] and the constraint free normal form [13]. These formalisms differ from each other not only in terms of the underlying constraint representation but also how these constraints are processed e.g., whether they require a constraint solver, splitting as needed versus shattering [15], etc.

The choice of the underlying constraint language can have a significant impact on the time as well as memory complexity of the inference procedure [15], and coming up with the right kind of constraint representation is of prime importance for the success of lifted inference techniques. Although, there

| Approach | Constraint Type | Constraint Aggregation | Tractable Solver | Lifting Algorithm |
|---|---|---|---|---|
| Lifted VE [4] | eq/ineq **no subset** | intersection **no union** | **no** | lifted VE |
| CFOVE [17] | eq/ineq **no subset** | intersection **no union** | yes | lifted VE |
| GCFOVE [25] | subset (tree-based) **no inequality** | intersection union | yes | lifted VE |
| Approx. LBP [24] | subset (hypercube) **no inequality** | intersection union | yes | lifted message passing |
| Knowledge Compilation (KC) [10, 7] | eq/ineq subset | intersection **no union** | **no** | first-order knowledge compilation |
| Lifted Inference from Other Side [13] | normal forms **(no constraints)** | **none** | yes | lifting rules: decomposer,binomial |
| PTP [11] | eq/ineq **no subset** | intersection **no union** | **no** | lifted search & sampling: decomposer, binomial |
| Current Work | eq/ineq subset | intersection union | yes | lifted search & sampling: decomposer,binomial single occurrence |

Table 1: A comparison of constraint languages proposed in literature across four dimensions. The deficiencies/missing properties for each language have been highlighted in bold. Among the existing work, only KC allows for a full set of constraints. GCFOVE (tree-based) and LBP (hypercubes) allow for subset constraints but they do not explicitly handle inequality. PTP does not handle subset constraints. For constraint aggregation, most approaches allow only intersection of atomic constraints. GCFOVE and LBP allow union of intersections (DNF) but only deal with subset constraints. See footnote 4 in Broeck [7] regarding KC. Lifted VE, KC and PTP use a general purpose constraint solver which may not be tractable. Our approach allows for all the features discussed above and uses a tractable solver. We propose a constrained solution for lifted search and sampling. Among earlier work, only PTP has looked at this problem (both search and sampling). However, it only allows a very restrictive set of constraints.

has been some work studying this problem in the context of lifted VE [25], lifted BP [24], and lifted knowledge compilation [10], existing literature lacks any systematic treatment of this issue in the context of lifted search and sampling based algorithms. This paper focuses on addressing this issue. Table 1 presents a detailed comparison of various constraint languages for lifted inference to date.

We make the following contributions. First, we propose a new constraint language called *setineq*, which allows for subset (i.e., allowed values are constrained to be either inside a subset or outside a subset), equality and inequality constraints (called *atomic constraints*) over substitutions of the variables. The set of allowed constraints is expressed as a union over individual constraint tuples, which in turn are conjunctions over atomic constraints. Our constraint language strictly subsumes several of the existing constraint representations and yet allows for efficient constraint processing, and more importantly does not require a separate constraint solver. Second, we extend the three main lifted inference rules: decomposer and binomial [13], and single occurrence [18] for MAP inference, to work with our proposed constraint language. We provide a detailed analysis of the lifted inference rules in our constraint formalism and formally prove that the normal form representation is strictly subsumed by our constraint formalism. Third, we show that evidence can be efficiently represented in our constraint formulation and is a key benefit of our approach. Specifically, based on the earlier work of Singla et al. [24], we provide an efficient (greedy) approach to convert the given evidence in the database tuple form to our constraint representation. Finally, we demonstrate experimentally that our new approach is superior to normal forms as well as many other existing approaches on several benchmark MLNs for both exact and approximate inference.

## 2 Markov Logic

We will use a strict subset of first order logic [22], which is composed of `constant`, `variable`, and `predicate` symbols. A term is a variable or a constant. A predicate represents a property of or relation between terms, and takes a finite number of terms as arguments. A `literal` is a predicate or its negation. A formula is recursively defined as follows: (1) a literal is a formula, (2) negation of a formula is a formula, (3) if $f_1$ and $f_2$ are formulas then applying binary logical operators such as $\wedge$ and $\vee$ to $f_1$ and $f_2$ yields a formula and (4) If $x$ is a variable in a formula $f$, then $\exists x \ f$ and $\forall x \ f$ are formulas. A first order theory (knowledge base (KB)) is a set of quantified formulas. We will restrict our attention to function-free finite first order logic theory with Herbrand interpretations [22], as done by most earlier work in this domain [5]. We will also restrict our

attention to the case of universally quantified variables. A ground atom is a predicate whose terms do not contain any variable in them. Similarly, a ground formula is a formula that has no variables. During the grounding of a theory, each formula is replaced by a conjunction over ground formulas obtained by substituting the universally quantified variables by constants appearing in the theory.

A Markov logic network (MLN) [5] (or a Markov logic theory) is defined as a set of pairs $\{f_i, w_i\}_{i=1}^m$ where $f_i$ is a first-order formula and $w_i$ is its weight, a real number. Given a finite set of constants $\mathcal{C}$, a Markov logic theory represents a Markov network that has one node for every ground atom in the theory and a feature for every ground formula. The probability distribution represented by the Markov network is given by $P(\theta) = \frac{1}{Z} \exp(\sum_{i=1}^m w_i n_i(\theta))$, where $n_i(\theta)$ denotes the number of true groundings of the $i^{th}$ formula under the assignment $\theta$ to the ground atoms (world) and $Z = \sum_{\theta'} \exp(\sum_{i=1}^m w_i * n_i(\theta'))$ is the normalization constant, called the *partition function*. It is well known that prototypical marginal inference task in MLNs – computing the marginal probability of a ground atom given evidence – can be reduced to computing the partition function [11]. Another key inference task is MAP inference in which the goal is to find an assignment to ground atoms that has the maximum probability.

In its standard form, a Markov logic theory is assumed to be constraint free i.e. all possible substitutions of variables by constants are considered during the grounding process. In this paper, we introduce the notion of a *constrained Markov logic* theory which is specified as a set of triplets $\{f_i, w_i, S_i^{\mathbf{x}}\}_{i=1}^m$ where $S_i^{\mathbf{x}}$ specifies a set (union) of constraints defined over the variables $\mathbf{x}$ appearing in the formula. During the grounding process, we restrict to those constant substitutions which satisfy the constraint set associated with a formula. The probability distribution is now defined using the restricted set of groundings allowed by the respective constraint sets over the formulas in the theory. Although, we focus on MLNs in this paper, our results can be easily generalized to other representations including weighted parfactors [3] and probabilistic knowledge bases [11].

## 3   Constraint Language

In this section, we formally define our constraint language and its canonical form. We also define two operators, join and project, for our language. The various features, operators, and properties of the constraint language presented this section will be useful when we formally extend various lifted inference rules to the constrained Markov logic theory in the next section (sec. 4).

**Language Specification.** For simplicity of exposition, we assume that all logical variables take values from the same domain $\mathcal{C}$. Let $\mathbf{x} = \{x_1, x_2, \ldots, x_k\}$ be a set of logical variables. Our constraint language called *setineq* contains three types of *atomic* constraints: (1) **Subset Constraints (setct)**, of the form $x_i \in \mathcal{C}$ (setinct), or $x_i \notin \mathcal{C}$ (setoutct); (2) **equality constraints (eqct)**, of the form $x_i = x_j$; and (3) **inequality constraints (ineqct)**, of the form $x_i \neq x_j$. We will denote an atomic constraint over set $\mathbf{x}$ by $A^{\mathbf{x}}$. A **constraint tuple** over $\mathbf{x}$, denoted by $T^{\mathbf{x}}$, is a conjunction of atomic constraints over $\mathbf{x}$, and a **constraint set** over $\mathbf{x}$, denoted by $S^{\mathbf{x}}$, is a disjunction of constraint tuples over $\mathbf{x}$. An example of a constraint set over a pair of variables $\mathbf{x} = \{x_1, x_2\}$ is $S^{\mathbf{x}} = T_1^{\mathbf{x}} \vee T_2^{\mathbf{x}}$, where $T_1^{\mathbf{x}} = [x_1 \in \{A, B\} \wedge x_1 \neq x_2 \wedge x_2 \in \{B, D\}]$, and $T_2^{\mathbf{x}} = [x_1 \notin \{A, B\} \wedge x_1 = x_2 \wedge x_2 \in \{B, D\}]$. An assignment $v$ to the variables in $\mathbf{x}$ is a **solution** of $T^{\mathbf{x}}$ if all constraints in $T^{\mathbf{x}}$ are satisfied by $v$. Since $S^{\mathbf{x}}$ is a disjunction, by definition, $v$ is also a solution of $S^{\mathbf{x}}$.

Next, we define a canonical representation for our constraint language. We require this definition because symmetries can be easily identified when constraints are expressed in this representation. We begin with some required definitions. The *support* of a subset constraint is the set of values in $\mathcal{C}$ that satisfies the constraint. Two subset constraints $A^{\mathbf{x_1}}$ and $A^{\mathbf{x_2}}$ are called *value identical* if $V_1 = V_2$, and *value disjoint* if $V_1 \cap V_2 = \phi$, where $V_1$ and $V_2$ are supports of $A^{\mathbf{x_1}}$ and $A^{\mathbf{x_2}}$ respectively. A constraint tuple $T^{\mathbf{x}}$ is transitive over equality if it contains the transitive closure of all its equality constraints. A constraint tuple $T^{\mathbf{x}}$ is transitive over inequality if for every constraint of the form $x_i = x_j$ in $T^{\mathbf{x}}$, whenever $T^{\mathbf{x}}$ contains $x_i \neq x_k$, it also contains $x_j \neq x_k$.

**Definition 3.1.** *A constraint tuple $T^{\mathbf{x}}$ is in **canonical form** if the following three conditions are satisfied: (1) for each variable $x_i \in \mathbf{x}$, there is exactly one subset constraint in $T^{\mathbf{x}}$, (2) all equality and inequality constraints in $T^{\mathbf{x}}$ are transitive and (3) all pairs of variables $x_1, x_2$ that participate either in an equality or an inequality constraint have identical supports. A constraint set $S^{\mathbf{x}}$ is in canonical form if all of its constituent constraint tuples are in canonical form.*

We can easily express a constraint set in an equivalent canonical form by enforcing the three conditions, one by one on each of its tuples. In our running example, $T_1^{\mathbf{x}}$ can be converted into canonical form by splitting it into four sets of constraint tuples $\{T_{11}^{\mathbf{x}}, T_{12}^{\mathbf{x}}, T_{13}^{\mathbf{x}}, T_{14}^{\mathbf{x}}\}$, where $T_{11}^{\mathbf{x}} = [x_1 \in \{B\} \wedge x_1 \neq x_2 \wedge x_2 \in \{B\}]$, $T_{12}^{\mathbf{x}} = [x_1 \in \{B\} \wedge x_2 \in \{D\}]$, $T_{13}^{\mathbf{x}} = [x_1 \in \{A\} \wedge x_2 \in \{B\}]$, and $T_{14}^{\mathbf{x}} = [x_1 \in \{A\} \wedge x_2 \in \{D\}]$. Similarly for $T_2^{\mathbf{x}}$. We include the conversion algorithm in the supplement due to lack of space. The following theorem summarizes its time complexity.

**Theorem 3.1.*** *Given a constraint set $S^{\mathbf{x}}$, each constraint tuple $T^{\mathbf{x}}$ in it can be converted to canonical form in time $O(mk + k^3)$ where $m$ is the total number of constants appearing in any of the subset constraints in $T^{\mathbf{x}}$ and $k$ is the number of variables in $\mathbf{x}$.*

We define following two operations in our constraint language.

**Join:** Join operation lets us combine a set of constraints (possibly defined over different sets of variables) into a single constraint. It will be useful when constructing formulas given constrained predicates (refer Section 4). Let $T^{\mathbf{x}}$ and $T^{\mathbf{y}}$ be constraints tuples over sets of variables $\mathbf{x}$ and $\mathbf{y}$, respectively, and let $\mathbf{z} = \mathbf{x} \cup \mathbf{y}$. The *join operation* written as $T^{\mathbf{x}} \bowtie T^{\mathbf{y}}$ results in a constraint tuple $T^{\mathbf{z}}$ which has the conjunction of all the constraints present in $T^{\mathbf{x}}$ and $T^{\mathbf{y}}$. Given the constraint tuple $T_1^{\mathbf{x}}$ in our running example and $T^{\mathbf{y}} = [x_1 \neq y \wedge y \in \{E, F\}]$, $T_1^{\mathbf{x}} \bowtie T^{\mathbf{y}}$ results in $[x_1 \in \{A, B\} \wedge x_1 \neq x_2 \wedge x_1 \neq y \wedge x_2 \in \{B, D\} \wedge y \in \{E, F\}]$. The complexity of join operation is linear in the size of constraint tuples being joined.

**Project:** Project operation lets us eliminate a variable from a given constraint tuple. This is key operation required in the application of Binomial rule (refer Section 4). Let $T^{\mathbf{x}}$ be a constraint tuple. Given $x_i \in \mathbf{x}$, let $\bar{\mathbf{x}}_i = \mathbf{x} \setminus \{x_i\}$. The *project operation* written as $\Pi_{\bar{\mathbf{x}}_i} T^{\mathbf{x}}$ results in a constraint tuple $T^{\bar{\mathbf{x}}_i}$ which contains those constraints in $T^{\mathbf{x}}$ not involving $x_i$. We refer to $T^{\bar{\mathbf{x}}_i}$ as the projected constraint for the variables $\bar{\mathbf{x}}_i$. Given a solution $\bar{\mathbf{x}}_i = \bar{\mathbf{v}}_i$ to $T^{\bar{\mathbf{x}}_i}$, the **extension count** for $\bar{\mathbf{v}}_i$ is defined as the number of unique assignments $x_i = v_i$ such that $\bar{\mathbf{x}}_i = \bar{\mathbf{v}}_i, x_i = v_i$ is a solution for $T^{\mathbf{x}}$. $T^{\bar{\mathbf{x}}_i}$ is said to be **count preserving** if each of its solutions has the same extension count. We require a tuple to be count preserving in order to correctly maintain the count of the number of solutions during the project operation (also refer Section 4.3).

**Lemma 3.1.*** *Let $T^{\mathbf{x}}$ be a constraint tuple in its canonical form. If $x_i \in \mathbf{x}$ is a variable which is either involved only in a subset constraint or is involved in at least one equality constraint then, the projected constraint $T^{\bar{\mathbf{x}}_i}$ is count preserving. In the former case, the extension count is given by the size of the support of $x_i$. In the latter case, it is equal to $1$.*

When dealing with inequality constraints, the extension count for each solution $\bar{\mathbf{v}}_i$ to the projected constraint $T^{\bar{\mathbf{x}}_i}$ may not be the same and we need to split the constraint first in order to apply the project operation. For example, consider the constraint $[x_1 \neq x_2 \wedge x_1 \neq x_3 \wedge x_1, x_2, x_3 \in \{A, B, C\}]$. Then, the extension count for the solution $x_2 = A, x_3 = B$ to the projected constraint $T^{\bar{\mathbf{x}}_1}$ is $1$ where extension count for the solution $x_2 = x_3 = A$ is $2$. In such cases, we need to split the tuple $T^{\mathbf{x}}$ into multiple constraints such that extension count property is preserved in each split. Let $\bar{\mathbf{x}}_i$ be a set of variables over which a constraint tuple $T^{\mathbf{x}}$ needs to be projected. Let $\mathbf{y} \subset \mathbf{x}$ be the set of variables with which $x_i$ is involved in an inequality constraint in $T^{\mathbf{x}}$. Then, tuple $T^{\mathbf{x}}$ can be broken into an equivalent constraint set by considering each possible division of $\mathbf{y}$ into a set of equivalence classes where variables in the same equivalence class are constrained to be equal and variables in different equivalence classes are constrained to be not equal to each other. The number of such divisions is given by the Bell number [15]. The divisions inconsistent with the already existing constraints over variables in $\mathbf{y}$ can be ignored. Projection operation has a linear time complexity once the extension count property has been ensured using splitting as described above (see the supplement for details).

## 4 Extending Lifted Inference Rules

We extend three key lifted inference rules: decomposer [13], binomial [13] and the single occurrence [18] (for MAP) to work with our constraint formulation. Exposition for Single Occurrence has been moved to supplement due to lack of space. We begin by describing some important definitions and assumptions. Let $M$ be a constrained MLN theory represented by a set of triplets $\{(f_i, w_i, S_i^{\mathbf{x}})\}_{i=1}^m$. We make three assumptions. First, we assume that each constraint set $S_i^{\mathbf{x}}$ is specified using setineq and is in canonical form. Second, we assume that each formula in the MLN is constant free. This can be achieved by replacing the appearance of a constant by a variable and introducing appropriate constraint over the new variable (e.g., replacing $A$ by a variable $x$ and a

constraint $x \in \{A\}$). Third, we assume that the variables have been standardized apart, i.e., each formula has a unique set of variables associated with it. In the following, $\mathbf{x}$ will denote the set of all the (logical) variables appearing in $M$. $\mathbf{x}_i$ will denote the set of variables in $f_i$. Similar to the work done earlier [13, 18], we divide the variables in a set of equivalence classes. Two variables are *Tied* to each other if they appear as the same argument of a predicate. We take the transitive closure of the Tied relation to obtain the variable equivalence classes. For example, given the theory: $P(x) \Rightarrow Q(x,y)$; $Q(u,v) \Rightarrow R(v)$; $R(w) \Rightarrow T(w,z)$, the variable equivalence classes are $\{x,u\}$, $\{y,v,w\}$ and $\{z\}$. We will use the notation $\hat{x}$ to denote the equivalence class to which $x$ belongs.

## 4.1 Motivation and Key Operations

The key intuition behind our approach is as follows. Let $x$ be a variable appearing in a formula $f_i$. Let $T^{\mathbf{x}_i}$ be an associated constraint tuple and $V$ denote the support for $x$ in $T^{\mathbf{x}_i}$. Then, since constraints are in canonical form, for any other variable $x' \in \mathbf{x}_i$ involved in (in)equality constraint with $x$ with $V'$ as the support, we have $V = V'$Therefore, every pair of values $v_i, v_j \in V$ behave identically with respect to the constraint tuple $T^{\mathbf{x}_i}$ and hence, are symmetric to each other. Now, we could extend this notion to other constraints in which $x$ appears provided the support sets $\{V_l\}_{l=1}^r$ of $x$ in all such constraints are either identical or disjoint. We could treat each support set $V_l$ for $x$ as a symmetric group of constants which could be argued about in unison. In an unconstrained theory, there is a single disjoint partition of constants i.e. the entire domain, such that the constants behave identically. Our approach generalizes this idea to a groups of constants which behave identically with each other. Towards this end, we define following 2 key operations over the theory which will be used over and again during application of lifted inference rules.

**Partitioning Operation:** We require the support sets of a variable (or sets of variables) over which lifted rule is being applied to be either identical or disjoint. We say that a theory $M$ defined over a set of (logical) variables $\mathbf{x}$ is **partitioned** with respect to the variables in the set $\mathbf{y} \subseteq \mathbf{x}$ if for every pair of subset constraints $A^{x_1}$ and $A^{x_2}$, $x_1, x_2 \in \mathbf{y}$ appearing in tuples of $S^{\mathbf{x}}$ the supports of $A^{x_1}$ and $A^{x_2}$ are either identical or disjoint (but not both). Given a partitioned theory with respect to variables $\mathbf{y}$, we use $\mathcal{V}^{\mathbf{y}} = \{V_l^{\mathbf{y}}\}_{l=1}^r$ to denote the set of various supports of variables in $\mathbf{y}$. We refer to the set $\mathcal{V}^{\mathbf{y}}$ as the partition of $\mathbf{y}$ values in $M$. Our partitioning algorithm considers all the support sets for variables in $\mathbf{y}$ and splits them such that all the splits are identical or disjoint. The constraint tuples can then be split and represented in terms of these fine-grained support sets. We refer the reader to the supplement section for a detailed description of our partitioning algorithm.

**Restriction Operation:** Once the values of a set of variables $\mathbf{y}$ have been partitioned into a set $\{V^{\mathbf{y}}\}_{l=1}^r$, while applying the lifted inference rules, we will often need to argue about those formula groundings which are obtained by restricting $\mathbf{y}$ values to those in a particular set $V_l^{\mathbf{y}}$ (since values in each such support set behave identically to each other). Given $x \in \mathbf{y}$, let $A_l^x$ denote a subset constraint over $x$ with $V_l^{\mathbf{y}}$ as its support. Given a formula $f_i$ we define its **restriction** to the set $V_l^{\mathbf{y}}$ as the formula obtained by replacing its associated constraint tuple $T^{\mathbf{x}_i}$ with a new constraint tuple of the form $T^{\mathbf{x}_i} \bigwedge_j A_l^{x_j}$ where the conjunction is taken over each variable $x_j \in \mathbf{y}$ which also appears in $f_i$. The restriction of an MLN $M$ to the set $V_l$, denoted by $M_l^{\mathbf{y}}$, is the MLN obtained by restricting each formula in $M$ to the set $V_l$. Restriction operation can be implemented in a straightforward manner by taking conjunction with the subset constraints having the desired support set for variables in $\mathbf{y}$. We next define the formulation of our lifting rules in a constrained theory.

## 4.2 Decomposer

Let $M$ be an MLN theory. Let $\mathbf{x}$ denote the set of variables appearing in $M$. Let $Z(M)$ denotes the partition function for $M$. We say that an equivalence class $\hat{x}$ is a decomposer [13] of $M$ if a) if $x \in \hat{x}$ occurs in a formula $f \in F$, then $x$ appears in every predicate in $f$ and b) If $x_i, x_j \in \hat{x}$, then $x_i, x_j$ do not appear as different arguments of any predicate $P$. Let $\hat{x}$ be a decomposer for $M$. Let $M'$ be a new theory in which the domain of all the variables belonging to equivalence class $\hat{x}$ has been reduced to a single constant. The decomposer rule [13] states that the partition function $Z(M)$ can be re-written using $Z(M')$ as $Z(M) = (Z(M'))^m$, where $m = |Dom(\hat{x})|$ in $M$. The proof follows from the fact that since $\hat{x}$ is a decomposer, the theory can be decomposed into $m$ independent but identical (up to the renaming of a constant) theories which do not share any random variables [13].

Next, we will extend the decomposer rule above to work with the constrained theories. We will assume that the theory has been partitioned with respect to the set of variables appearing in the

decomposer $\hat{x}$. Let the partition of $\hat{x}$ values in $M$ be given by $\mathcal{V}^{\hat{x}} = \{V_l^{\hat{x}}\}_{l=1}^r$. Now, we define the decomposer rule for a constrained theory using the following theorem.

**Theorem 4.1.** * *Let $M$ be a partitioned theory with respect to the decomposer $\hat{x}$. Let $M_l^{\hat{x}}$ denote the restriction of $M$ to the partition element $V_l^{\hat{x}}$. Let $M_l'^{\hat{x}}$ further restricts $M_l^{\hat{x}}$ to a singleton $\{v\}$ where $v \in V^{\hat{x}}$ is some element in the set $V^{\hat{x}}$. Then, the partition function $Z(M)$ can be written as $Z(M) = \Pi_{l=1}^r Z(M_l^{\hat{x}}) = \Pi_{l=1}^r Z(M_l'^{\hat{x}})^{|V_l^{\hat{x}}|}$*

### 4.3 Binomial

Let $M$ be an unconstrained MLN theory and $P$ be a unary predicate. Let $\mathbf{x_j}$ denote the set of variables appearing as first argument of $P$. Let $Dom(x_j) = \{c_i\}_{i=1}^n, \forall x_j \in \mathbf{x_j}$. Let $M_k^P$ be the theory obtained from $M$ as follows. Given a formula $f_i$ with weight $w_i$ in which $P$ appears, wlog let $x_j$ denote the argument of $P$ in $f_i$. Then, for every such formula $f_i$, we replace it by two new formulas, $f_i^t$ and $f_i^f$, obtained by a) substituting $true$ and $false$ for the occurrence of $P(x_j)$ in $f_i$, respectively, and b) when $x_j$ occurs in $f_i^t$ or $f_i^f$, reducing the domain of $x_j$ to $\{c_i\}_{i=1}^k$ in $f_i^t$ and $\{c_i\}_{i=k+1}^n$ in $f_i^f$ where $n = |Dom(x_j)|$. The weight $w_i^t$ of $f_i^t$ is equal to $w_i$ if it has an occurrence of $x_j$, $w_i * k$ otherwise. Similarly, for $f_i^f$. The Binomial rule [13] states that the partition function $Z(M)$ can be written as: $Z(M) = \sum_{k=0}^n \binom{n}{k} \left( Z(M_k^P) \right)$. The proof follows from the fact that calculation of $Z$ can be divided into $n + 1$ cases, where each case corresponds to considering $\binom{n}{k}$ equivalent possibilities for $k$ number of $P$ groundings being true and $n - k$ being false, $k$ ranging from 0 to $n$.

Next, we extend the above rule for a constrained theory $M$. Let $P$ be singleton predicate and $\mathbf{x_j}$ be set of variables appearing as first arguments of $P$ as before. Let $M$ be partitioned with respect to $\mathbf{x_j}$ and $\mathcal{V}^{\mathbf{x_j}} = \{V_l^{\mathbf{x_j}}\}_{l=1}^r$ denote the partition of $\mathbf{x_j}$ values in $M$. Let $F^P$ denote the set of formulas in which $P$ appears. For every formula $f_i \in F^P$ in which $x_j$ appears only in $P(x_j)$, assume that the projections over the set $\bar{\mathbf{x_j}}$ are count preserving. Then, we obtain a new MLN $M_{l,k}^P$ from $M$ in the following manner. Given a formula $f_i \in F^P$ with weight $w_i$ in which $P$ appears, do the following steps 1) restrict $f_i$ to the set of values $\{v | v \notin V_l^{\mathbf{x_j}}\}$ for variable $x_j$ 2) for the remaining tuples (i.e. where $x_j$ takes the values from the set $V_l^{\mathbf{x_j}}$), create two new formulas $f_i^t$ and $f_i^f$ obtained by restricting $f_i^t$ to the set $\{V_{l_1}^{\mathbf{x_j}}, \ldots V_{l_k}^{\mathbf{x_j}}\}$ and $f_i^f$ to the set $\{V_{l_{k+1}}^{\mathbf{x_j}}, \ldots, V_{l_{n_l}}^{\mathbf{x_j}}\}$, respectively. Here, the subscript $n_l = |V_l^{\mathbf{x_j}}|$ 3) Canonicalize the constraints in $f_i^t$ and $f_i^f$ 4) Substitute $true$ and $false$ for $P$ in $f_i^t$ and $f_i^f$ respectively 5) If $x_j$ appears in $f_i^t$ (after the substitution), its weight $w_i^t$ is equal to $w_i$, otherwise split $f_i^t$ into $\{f_{i_d}^t\}_{d=1}^D$ such that projection over $\bar{\mathbf{x_j}}$ in each tuple of $f_{i_d}^t$ is *count preserving* with extension count given by $e_{l_d}^t$. The weight of each $f_{i_d}^t$ is $w_i * e_{l_d}^t$. Similarly, for $f_i^f$. We are now ready to define the Binomial formulation for a constrained theory:

**Theorem 4.2.** * *Let $M$ be an MLN theory partitioned with respect to variable $x_j$. Let $P(x_j)$ be a singleton predicate. Let the projections $T^{\bar{\mathbf{x_j}}}$ of tuples associated with the formulas in which $x_j$ appears only in $P(x_j)$ be count preserving. Let $\mathcal{V}^{x_j} = \{V_l^{x_j}\}_{l=1}^r$ denote the partition of $x_j$ values in $M$ and let $n_l = |V_l^{x_j}|$. Then, the partition function $Z(M)$ can be computed using the recursive application of the following rule for each $l$:*

$$Z(M) = \sum_{k=0}^{n_l} \binom{n_l}{k} \left( Z(M_{l,k}^P) \right)$$

We apply Theorem 4.2 recursively for each partition component in turn to eliminate $P(x_j)$ completely from the theory. The Binomial application as described above involves $\prod_{l=1}^r (n_l + 1)$ computations of $Z$ whereas a direct grounding method would involve $2^{\sum_l n_l}$ computations (two possibilities for each grounding of $P(x_j)$ in turn). See the supplement for an example.

### 4.4 Normal Forms and Evidence Processing

**Normal Forms:** Normal form representation [13] is an unconstrained representation which requires that a) there are no constants in any formula $f_l \in F$ b) the domain of variables belonging to an equivalence class $\hat{x}$ are identical to each other. An (unconstrained) MLN theory with evidence can be converted into normal form by a series of mechanical operations in time polynomial in the size

| Domain | Source | Rules | Type (# of const.) | Evidence |
|--------|--------|-------|--------------------|----------|
| Friends & Smokers (FS) | Alchemy [5] | Smokes($p$) $\Rightarrow$ Cancer($p$); Smokes($p1$) $\wedge$ Friends($p1,p2$) $\Rightarrow$ Smokes($p2$) | person (**var**) | Smokes Cancer |
| WebKB | Alchemy [25],[24] | PageClass($p1,+c1$) $\wedge$ PageClass($p2,+c2$) $\Rightarrow$ Links($p1,p2$) | page (271) class (5) | PageClass |
| IMDB | Alchemy [16] | Director($p$) $\Rightarrow$ !WorksFor($p1,p2$) Actor($p$) $\Rightarrow$ !Director($p$); Movie($m,p1$) $\wedge$ WorksFor($p1,p2$) $\Rightarrow$ Movie($m,p2$) | person(278) movie (20) | Actor Director Movie |

Table 2: Dataset Details. **var**: domain size varied. '+': a separate weight learned for each grounding

of the theory and the evidence [13, 18]. Any variable values appearing as a constant in a formula or in evidence is split apart from the rest of the domain and a new variable with singleton domain created for them. Constrained theories can be normalized in a similar manner by 1) splitting apart those variables appearing any subset constraints. 2) simple variable substitution for equality and 3) introducing explicit evidence predicates for inequality. We can now state the following theorem.

**Theorem 4.3.** * *Let $M$ be a constrained MLN theory. The application of the modified lifting rules over this constrained theory can be exponentially more efficient than first converting the theory in the normal form and then applying the original formulation of the lifting rules.*

**Evidence Processing:** Given a predicate $P_j(x_1, \ldots, x_k)$ let $E_j$ denote its associated evidence. Further, $E_j^t$ ($E_j^f$) denote the set of ground atoms of $P_j$ which are assigned $true$ ($false$) in evidence. Let $E_j^u$ denote the set of groundings which are unknown (neither $true$ nor $false$.) Note that the set $E_j^u$ is implicitly specified. The first step in processing evidence is to convert the sets $E_j^t$ and $E_j^f$ into the constraint representation form for every predicate $P_j$. This is done by using the hypercube representation [24] over the set of variables appearing in predicate $P_j$. A hypercube over a set of variables can be seen as a constraint tuple specifying a subset constraint over each variable in the set. A union of hypercubes represents a constraint set representing the union of corresponding constraint tuples. Finding a minimal hypercube decomposition in NP-hard and we employ the greedy top-down hypercube construction algorithm as proposed Singla et al. [24] (Algorithm 2). The constraint representation for the implicit set $E_j^u$ can be obtained by eliminating the set $E_j^t \cup E_j^f$ from its bounding hypercube (i.e. one which includes all the groundings in the set) and then calling the hypercube construction algorithm over the remaining set. Once the constraint representation has been created for every set of evidence (and non-evidence) atoms, we join them together to obtain the constrained representation. The join over constraints is implemented as described in Section 3.

## 5 Experiments

In our experiments, we compared the performance of our constrained formulation of lifting rules with the normal forms for the task of calculating the partition function $Z$. We refer to our approach as *SetInEq* and normal forms as *Normal*. We also compared with PTP [11] available in Alchemy 2 and GCFVOE [25] system. [1] Both our systems and GCFOVE are implemented in Java. PTP is implemented in C++. We experimented on four benchmark MLN domains for calculating the partition function using exact as well as approximate inference. Table 2 shows the details of our datasets. Details for one of the domains Professor and Students (PS) [11] are presented in supplement due to lack of space. Evidence was the only type of constraint considered in our experiments. The experiments on all the datasets except WebKB were carried on a machine with 2.20GHz Intel Core i3 CPU and 4GB RAM. WebKB is a much larger dataset and we ran the experiments on 2.20 GHz Xeon(R) E5-2660 v2 server with 10 cores and 128 GB RAM.

### 5.1 Exact Inference

We compared the performance of the various algorithms using exact inference on two of the domains: FS and PS. We do not compare the value of $Z$ since we are dealing with exact inference In the following, $r\%$ evidence on a type means that $r\%$ of the constants of the type are randomly selected and evidence predicate groundings in which these constants appear are randomly set to true or false. Remaining evidence groundings are set to unknown. $y$-axis is plotted on log scale in the following 3 graphs. Figure 1a shows the results as the domain size of $person$ is varied from 100 to 800 with $40\%$ evidence in the FS domain. We timed out an algorithm after 1 hour. PTP failed to

scale to even 100 size and are not shown in the figure. The time taken by Normal grows very fast and it times out after 500 size. SetInEq and GCFOVE have a much slower growth rate. SetInEq is about an order of magnitude faster than GCFVOE on all domain sizes. Figure 1b shows the time taken by the three algorithms as we vary the evidence on $person$ with a fixed domain size of 500. For all the algorithms, the time first increases with evidence and then drops. SetInEq is up to an order of magnitude faster than GCFVOE and upto 3 orders of magnitude faster than Normal. Figure 1c plots the number of nodes expanded by Normal and SetInEQ. GCFOVE code did not provide any such equivalent value. As expected, we see much larger growth rate for Normal compared to SetInEq.

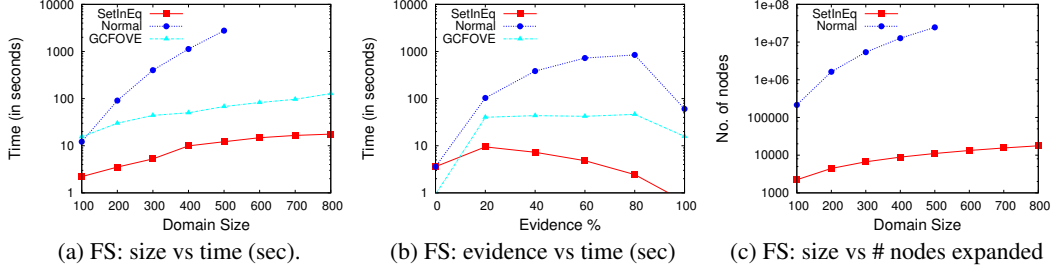

(a) FS: size vs time (sec).　　　(b) FS: evidence vs time (sec)　　　(c) FS: size vs # nodes expanded

Figure 1: Results for exact inference on FS

## 5.2 Approximate Inference

For approximate inference, we could only compare Normal with SetInEq. GCFOVE does not have an approximate variant for computing marginals or partition function. PTP using importance sampling is not fully implemented in Alchemy 2. For approximate inference in both Normal and SetInEq, we used the unbiased importance sampling scheme as described by Gogate & Domingos [11]. We collected a total of 1000 samples for each estimate and averaged the $Z$ values. In all our experiments below, the $log(Z)$ values calculated by the two algorithms were within $1\%$ of each other hence, the estimates are comparable with other. We compared the performance of the two algorithms on two real world datasets IMDB and WebKB (see Table 2). For WebKB, we experimented with 5 most frequent page classes in Univ. of Texas fold. It had close to 2.5 million ground clauses. IMDB has 5 equal sized folds with close to $15K$ groundings in each. The results presented are averaged over the folds. Figure 2a ($y$-axis on log scale) shows the time taken by two algorithms as we vary the subset of pages in our data from 0 to 270. The scaling behavior is similar to as observed earlier for datasets. Figure 2b plots the timing of the two algorithms as we vary the evidence % on IMDB. SetInEq is able to exploit symmetries with increasing evidence whereas Normal's performance degrades.

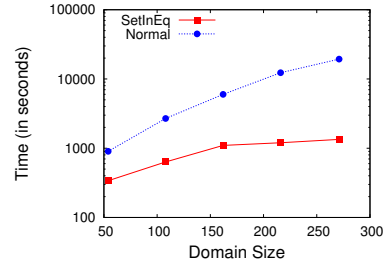

(a) WebKB: size vs time (sec)

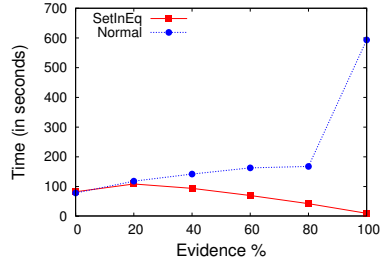

(b) IMDB: evidence % vs time (sec)

Figure 2: Results using approximate inference on WebKB and IMDB

## 6 Conclusion and Future work

In this paper, we proposed a new constraint language called *SetInEq* for relational probabilistic models. Our constraint formalism subsumes most existing formalisms. We defined efficient operations over our language using a canonical form representation and extended 3 key lifting rules i.e., decomposer, binomial and single occurrence to work with our constraint formalism. Experiments on benchmark MLNs validate the efficacy of our approach. Directions for future work include exploiting our constraint formalism to facilitate approximate lifting of the theory.

## 7 Acknowledgements

Happy Mittal was supported by TCS Research Scholar Program. Vibhav Gogate was partially supported by the DARPA Probabilistic Programming for Advanced Machine Learning Program under AFRL prime contract number FA8750-14-C-0005. Parag Singla is being supported by Google travel grant to attend the conference. We thank Somdeb Sarkhel for helpful discussions.

## Footnotes

[1]Alchemy-2:code.google.com/p-alchemy-2,GCFOVE: https:dtai.cs.kuleuven.be/software/gcfove

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
