[Supplementary Material]

# Supplementary Material: Lifted Inference Rules with Constraints

**Happy Mittal, Anuj Mahajan**
Dept. of Comp. Sci. & Engg.
I.I.T. Delhi, Hauz Khas
New Delhi, 110016, India
happy.mittal@cse.iitd.ac.in,
anujmahajan.iitd@gmail.com

**Vibhav Gogate**
Dept. of Comp. Sci.
Univ. of Texas Dallas
Richardson, TX 75080, USA
vgogate@hlt.utdallas.edu

**Parag Singla**
Dept. of Comp. Sci. & Engg.
I.I.T. Delhi, Hauz Khas
New Delhi, 110016, India
parags@cse.iitd.ac.in

## 3 Constraint Language

Any constraint tuple $T^{\mathbf{x}}$ can converted into its canonical form. Following lemmas describe how a constraint tuple can be converted into canonical form with respect to each type of constraint.

**Lemma. 1** *Given a constraint tuple $T^{\mathbf{x}}$, it can be equivalently written in canonical form with respect to subset constraints over any given variable $x \in \mathbf{x}$.*

*Proof.* To convert $T^{\mathbf{x}}$ into the canonical form with respect to variable $x$, we need to find an atomic constraint $A^x$ which can equivalently represent all the subset constraints over $x$ in $T^{\mathbf{x}}$. Such an $A^x$ can be obtained by enforcing that $x$ belongs to the set difference of a) intersection of the subsets $\mathcal{C}_i$'s such that there is a constraint of the form $x \in \mathcal{C}_i$ in $T^{\mathbf{x}}$ And b) union of the subsets $\mathcal{C}_j$'s such that there is a constraint of the form $x \notin \mathcal{C}_j$ in $T^{\mathbf{x}}$. In other words, $A^x$ can be written as $x \in (\cap_i C_i) \setminus (\cup_j C_j)$ where $(x \in C_i) \in T^{\mathbf{x}} \ \forall i$ and $(x \notin C_j) \in T^{\mathbf{x}} \ \forall j$. $\qquad \square$

### 3.1 Splitting Operation

We define the operation $split(A^{x_1}, A^{x_2})$ which inputs the subset constraints $A^{x_1}$ and $A^{x_2}$ and splits the input constraints into sets of constraints $A^{x_1}.split = \{A_I^{x_1}, A_D^{x_1}\}$ and $A^{x_2}.split = \{A_I^{x_2}, A_D^{x_2}\}$, which are all value identical or value disjoint with each other. If $V_1$ and $V_2$ are the supports for $A^{x_1}$ and $A^{x_2}$, respectively, and if, $V_{1I}, V_{1D}$ are the supports for $A_I^{x_1}$ and $A_D^{x_1}$, respectively, then $V_{1I} = V_1 \cap V_2$ and $V_{1D} = V_1 \setminus V_2$ It is easy to see that $V_1 = V_{1I} \cup V_{1D}$. In other words, the union of the constraints in the split is equivalent to the original constraint. Similarly, for the split $\{A_I^{x_2}, A_D^{x_2}\}$, and the corresponding supports, $V_{2D}, V_{2D}$, respectively. $split$ operation shown in Algorithm 1 can be implemented by considering the four cases described in Table 1, one for each combination of the forms $A^{x_1}$ and $A^{x_2}$ can take. The table lists only the splits for $A^{x_1}$. The splits for $A^{x_2}$ can be obtained from the same table by realizing that $split$ operation is symmetric.

---

**Algorithm 1** Split Constraints

---

**split**$(A^{x_1}, A^{x_2})$
   $A^{x_1}.split = \{A_I^{x_1}, A_D^{x_2}\}$ (using Table 1)
   $A^{x_2}.split = \{A_I^{x_2}, A_D^{x_2}\}$ (using symmetric analogue for $x_2$)
   **return**$(A^{x_1}.split, A^{x_2}.split)$

---

**Lemma. 2** *Let $T^{\mathbf{x}}$ be a constraint tuple over $\mathbf{x}$. Let $x_1$ and $x_2$ be two variables in $\mathbf{x}$ involved in the constraint $x_1 = x_2$. Then, $T^{\mathbf{x}}$ can be equivalently written in its canonical form with respect to the equality constraint over $x_1$ and $x_2$.*

*Proof.* Let $T^{\mathbf{x}}$ be in its canonical form with respect to its subset constraints. Let $A^{x_1}$ and $A^{x_2}$ be the subset constraints over variables $x_1$ and $x_2$ respectively in $T^{\mathbf{x}}$. Since we have a constraint $x_1 = x_2$, the values of variables $x_1$ and $x_2$ must be equal in any solution of the constraint and hence, we can equivalently restrict the supports of $x_1$ and $x_2$ (i.e. those allowed by subset constraints) to be the

| # | $A^{x_1}$ | $A^{x_2}$ | $A_I^{x_1}$ (Split $I$) | $A_D^{x_1}$ (Split $D$) |
|---|-----------|-----------|--------------------------|--------------------------|
| 1 | $x_1 \in C_1$ | $x_2 \in C_2$ | $x_1 \in C_1 \cap C_2$ | $x_1 \in C_1 \setminus C_2$ |
| 2 | $x_1 \in C_1$ | $x_2 \notin C_2$ | $x_1 \in C_1 \setminus C_2$ | $x_1 \in C_1 \cap C_2$ |
| 3 | $x_1 \notin C_1$ | $x_2 \in C_2$ | $x_1 \in C_2 \setminus C_1$ | $x_1 \notin C_1 \cup C_2$ |
| 4 | $x_1 \notin C_1$ | $x_2 \notin C_2$ | $x_1 \notin C_1 \cup C_2$ | $x_1 \in C_2 \setminus C_1$ |

Table 1: Split of subset constraint over $x_1$

set $V_1 \cap V_2$ where $V_1$, $V_2$ are the supports of $x_1$ and $x_2$ in the original tuple. Consider the result of $split(A^{x_1}, A^{x_2})$ over constraints $A^{x_1}$ and $A^{x_2}$, and let $A_I^{x_1}$ and $A_I^{x_2}$ be the splits for $A^{x_1}$ and $A^{x_2}$, respectively, returned by the operation corresponding to the intersection $V_1 \cap V_2$. Then, the desired tuple $T_I^{\mathbf{x}}$ in the canonical form can be obtained by having all the constraints in $T_I^{\mathbf{x}}$ as in $T^{\mathbf{x}}$ but replacing $A^{x_1}$ and $A^{x_2}$ by $A_I^{x_1}$ and $A_I^{x_2}$, respectively. □

**Lemma. 3** *Let $T^{\mathbf{x}}$ be a constraint tuple over $\mathbf{x}$. Let $x_1 \neq x_2$ be an inequality constraint in $T^{\mathbf{x}}$. Then, $T^{\mathbf{x}}$ can be broken into an equivalent constraint set $S^{\mathbf{x}} = \bigwedge_{i=1}^{k} T_i^{\mathbf{x}}$ such that a) $T_1^{\mathbf{x}}$ retains the inequality constraint and it is canonical form with respect to the inequality constraint $x_1 \neq x_2$ b) $T_i^{\mathbf{x}}, \forall i > 1$ does not contain the inequality constraint.*

*Proof.* As earlier, let $T^{\mathbf{x}}$ be already converted into its canonical form with respect to its subset constraints on $x_1$ and $x_2$ using lemma 3. Let the $A^{x_1}$ and $A^{x_2}$ be the subset constraints for $x_1$ and $x_2$, respectively, and let $V_1$ and $V_2$ be the respective supports. Then, we apply the operation $split(A^{x_1}, A^{x_2})$ over $A^{x_1}$ and $A^{x_2}$, and let $\{A_I^{x_1}, A_D^{x_1}\}$ and $\{A_I^{x_2}, A_D^{x_2}\}$ be the resulting splits. Then, we construct the constraint set $S^{\mathbf{x}}$ containing 4 constraint tuple $\{T^{\mathbf{x}}\}_{i=1}^{4}$ as follows. $T_i^{\mathbf{x}}$'s inherit all the constraints in $T^{\mathbf{x}}$ except $A^{x_1}$ and $A^{x_2}$. The inequality constraint $x_1 \neq x_2$ is present only in the tuple $T_1^{x}$ (and absent in others). In $T_1^{x}$, we replace the constraints $A^{x_1}$ and $A^{x_2}$ by $A_I^{x_1}$ and $A_I^{x_2}$, respectively. The supports for both $x_1$ and $x_2$ in $T_I^{x_1}$ are given by the intersection $V_1 \cap V_2$. Hence, $T_1^{\mathbf{x}}$ is in canonical form with respect to the inequality constraint. The remaining 3 tuples correspond to choosing one of the constraints from $\{A_I^{x_1}, A_D^{x_2}\}$ for $x_1$ and one from the set $\{A_I^{x_2}, A_D^{x_2}\}$ for $x_2$ (and leaving out $(A_I^{\mathbf{x_1}}, A_I^{\mathbf{x_2}})$ combination which has already been covered). Intuitively, the 4 tuples divide the supports of $x_1$, $x_2$ into 4 possible cases depending on whether the constant subset is allowed by both the variables (intersection of supports) or only one of the variables and not the other (set difference of supports). Hence, all possible combinations of $V_1$ and $V_2$ are covered by these four tuples, resulting an equivalent constraint set (remaining constraints are the same as in $T^{\mathbf{x}}$). $T_1^{\mathbf{x}}$ is the only tuple allowing for common values of $x_1$ and $x_2$, so we can safely omit the inequality constraint from the other tuples. □

**Theorem 3.1.** *A constraint tuple $T^{\mathbf{x}}$ can be converted into its canonical form in time $O(mk + k^3)$ where $m$ is the total number of constants appearing in any of the subset constraints in $T^{\mathbf{x}}$ and $k$ is the number of variables in $\mathbf{x}$.*

*Proof.* Subset constraints can be processed using the proof of lemma 1. The procedure involves a constant number of set operations, once for each variable, which can all be implemented in time linear in the number of constants in the tuple (using a hashtable). The multiple variable constraints can again be taken care of using proof for lemma 3. There is a total of $O(k^2)$ such constraints and each involves a constant number of set operations over constants in the tuple. A closure for transitivity can be obtained as follows. Each variable is assigned a unique identifier in the beginning. Each time an equality constraint is processed, the lower of the identifiers of the variables in the constraint is taken and assigned to all the variables which share the identifier with any of the variables in the constraint. There is a total of $O(k^2)$ equality constraints and each re-assignment takes $O(k)$ time. The operation is independent of the number of constants in $T^{\mathbf{x}}$. □

**Lemma 3.1.** *Let $T^{\mathbf{x}}$ be a constraint tuple in its canonical form. If $x_i \in \mathbf{x}$ is a variable involved only in subset constraint or involved in at least one equality constraint then, every solution $\bar{\mathbf{v}}_i$ to the projected constraint $T^{\mathbf{x}_i}$ has the same extension count. In the former case, the extension count is the size of the support of $x_i$. In the latter case, it is equal to 1.*

*Proof.* When $x_i \in \mathbf{x}$ is involved only in subset constraint, it is not involved in any constraint with another variable, so the values it can take in a solution are independent of other variable assignments. Every solution $\bar{\mathbf{v}}_\mathbf{i}$ over $T^{\mathbf{x}_\mathbf{i}}$ can be extended to $T^\mathbf{x}$ by choosing any value from $V_i$, the support of $x_i$, and hence the same extension count of $|V_i|$.

Let $x_i$ is involved in an equality constraint with some variable $x_j$. Since the tuple is in canonical form with respect to equality constraints, $V_i = V_j$ where $V_i, V_j$ are the supports for $x_i, x_j$, respectively. Further, if $x_k$ is another variable involved in (in)equality constraint with $x_i$ in $T^\mathbf{x}$, it is also involved in the corresponding (in)equality constraint with $x_j$ by transitivity property of canonical forms. This constraint involving $x_j, x_k$ must be present in $T^{\mathbf{x}_\mathbf{i}}$ by the definition of a projection. Therefore, any admissible solution $\bar{\mathbf{v}}_\mathbf{i}$ over $T^{\mathbf{x}_\mathbf{i}}$ can be extended to $T^\mathbf{x}$ by choosing a value $v_i \in V_i$ such that $v_i = v_j$ [1], and this is the only valid extension. Hence, every solution $\bar{\mathbf{v}}_\mathbf{i}$ to the projected constraint $T^{\mathbf{x}_\mathbf{i}}$ has the same extension count of 1. □

Next, we describe the time complexity of the project operation. Suppose the extension count property has been ensured by splitting the constraint as described earlier in the text. Let $k$ be the number of the variables involved in the inequality constraints. Then, the number of splits is $O(\text{BellNumber}(k))$ [2]. For each split, the complexity of the project operation is linear in the size of the constraint. Therefore, the total complexity of the operation is given by $O(l * \text{BellNumber}(k))$, where $l$ is the size of the original constraint. When dealing with models such as MLNs, $k$ is at most the number of variables in a formula, which is typically very small (single digit).

## 4 Lifted Inference Rules

Algorithm 3 gives the pseudocode for partitioning a theory $M$ with respect to a set of variables $\mathbf{y}$. In Algorithm 3, $S_M^\mathbf{x}$ represents the set of all the constraints in $M$ (line 2). $\mathcal{A}_M^\mathbf{y}$ represents the set of subset constraints in $M$ over $\mathbf{y}$ variables (line 3). This set is split into equivalent non-intersecting sets of constraints (line 4). This can be done by Algorithm 2. The resulting set of constraints can be thought of as the *base set* from which all the original constraints over $\mathbf{y}$ can be constructed. For every constraint tuple $T^\mathbf{x}$ in every formula for every variable $y_j \in \mathbf{y}$ (lines 5-8). we extract the subset constraints $A^{y_j}$ (line 9), find the constraints in the base set which represent $A^{y_j}$ (i.e. have non-empty intersection with $A^{y_j}$) (line 10) and then replace the tuple $T^\mathbf{x}$ by a set of tuples one for each element in the base set elements for $A^{y_j}$ (line 11). The original set of constraint tuples is replaced with the new (equivalent) set of tuples at the end of the iteration (line 13). The process is complete once iterate over all the variables $y_j \in \mathbf{y}$.

To split the set of subset constraints into equivalent non-intersecting sets of constraints, Algorithm 2 creates a $bitStr$ (a vector of bits) for each value $v$. $bitStr$ stores a bit for every constraint which is set if $v$ appears in that constraint. $bitStrToVals$ is a mapping from bit vectors to sets of values. Intuitively, each element of $bitStrToVals$ maps a bit vector to a set of values having the corresponding bit vector (this mapping can be constructed using standard hashing techniques). Therefore, $bitStrToVals$ enforces the desired partition. Finally $getSubsetConstraints$ returns the subset constraints corresponding to partitions created in $bitStrToVals$.

---

**Algorithm 2** Split into non-intersecting constraints

---

**SplitNonIntersect**($\mathcal{A}_M^\mathbf{y}$)
    $V \leftarrow \bigcup_{y_i \in y} V^{y_i}$ // $V^{y_i}$ is support set of $\mathcal{A}_M^{\mathbf{y}_\mathbf{i}}$
    $bitStrToVals = \{\}$ //empty mapping
    **for** $v \in V$ **do**
        $bitStr = \text{getBitVec}(v, \mathcal{A}_M^\mathbf{y})$
        $bitStrToVals[bitStr]$.add($v$)
    **end for**
    return $getSubsetConstraints(bitStrToVals)$

---

Next, we describe the time complexities of the partition and the restriction operations. Let $n$ be the total number of constants appearing in $m$ subset constraints. Then assuming constant time hashing, the time complexity of the partition operation is $O(mn)$. Let $y$ denote the set of variables being restricted. Let $m$ denote the total number of constraints in which $y$ variables appear. Let $n$ be the

**Algorithm 3** Partition the theory with respect to a given set of variables

---

1: **partition**$(M, \mathbf{y})$
2:    $S_M^{\mathbf{x}} \leftarrow \cup_{l=1}^m S_l^{\mathbf{x}}$
3:    $\mathcal{A}_M^{\mathbf{y}} \leftarrow getAllSubsetCt(S_M^{\mathbf{x}}, \mathbf{y})$ ;
4:    $\mathcal{A}_M^{\mathbf{y}} \leftarrow SplitNonIntersect(\mathcal{A}_M^{\mathbf{y}})$;
5: **for all** $y_j \in \mathbf{y}$ **do**
6:      **for all** $l \in \{1, \ldots m\}$ **do**
7:        $S_l'^{\mathbf{x}} \leftarrow \{\}$
8:        **for all** $T^{\mathbf{x}} \in S_l^{\mathbf{x}}$ **do**
9:          $A^{y_j} = T^{\mathbf{x}}.getSubsetCt(y_j)$
10:          $\mathcal{A}^{y_j} \leftarrow getIntersecting(A^{y_j}, \mathcal{A}_M^{\mathbf{y}})$;
11:          $S_l^{\mathbf{x}} \leftarrow S_l'^{\mathbf{x}} \cup replace(T^{\mathbf{x}}, y_j, A^{y_j}, \mathcal{A}^{y_j})$
12:        **end for**
13:        $S_l^{\mathbf{x}} \leftarrow S_l'^{\mathbf{x}}$
14:      **end for**
15: **end for**

---

number of constants appearing in those constraints. The complexity of the restriction operation is $O(|y| * m * |V|)$ where $|V|$ is the size of the support set for the restriction.

**Theorem 4.1.** *Let $M$ be a partitioned theory with respect to the decomposer $\hat{x}$. Let $M_l^{\hat{x}}$ denote the restriction of $M$ to the partition element $V_l^{\hat{x}}$. Let $A^x$ be a constraint with $V_l^{\hat{x}}$ as the set of potential values. Let $M_l'^{\hat{x}}$ further restricts $M_l^{\hat{x}}$ to a singleton for $A^x$. Then, the partition function $Z(M)$ can be written as $Z(M) = \Pi_{l=1}^r Z(M_l^{\hat{x}}) = \Pi_{l=1}^r Z(M_l'^{\hat{x}})^{|V_l^{\hat{x}}|}$*

*Proof.* Since $\hat{x}$ is decomposer, it is easy to see that restriction of $M$ to the partition elements for values of $\hat{x}$ breaks $M$ into $r$ independent components. Each grounding of $M$ is covered in some restriction since we consider restriction to every possible partition element. Hence, the partition function for $M$ can be written as the product of partition functions for each of the $r$ restrictions. Now, the next thing remaining to show is that the partition function $Z(M_l^{\hat{x}})$ for each of the restrictions is equal to the expression $Z(M_l'^{\hat{x}})^{|V_l^{\hat{x}}|}$.

For each of the restriction of the form $M_l^{\hat{x}}$, the values in the set $V_l^{\hat{x}}$ are symmetrical to each other due to the partitioning of the theory. Further, since $\hat{x}$ is a decomposer, groundings corresponding to each value in the set $V_l^{\hat{X}}$ are disjoint with each other i.e. do not share atoms. Hence, the partition function for $M_l^{\hat{x}}$ can be written as the product of partition functions for $|V_l^{\hat{x}}|$ independent MLNs which are identical to each other (upto renaming of a constant). These identical MLNs are described by the restriction $M_l'^{\hat{x}}$ of $M_l^{\hat{x}}$ to the singleton $A^x$ as define above. Hence, the result follows. $\square$

**Theorem 4.2.** *Let $M$ be an MLN theory partitioned with respect to variable $x_j$. Let $P(x_j)$ be a singleton predicate. Let the projections $T^{\mathbf{x_j}}$ of tuples associated with the formulas in which $x_j$ appears only in $P(x_j)$ be count preserving. Let $\mathcal{V}^{x_j} = \{V_l^{x_j}\}_{l=1}^r$ denote the partition of $x_j$ values in $M$ and let $n_l = |V_l^{x_j}|$. Then, the partition function $Z(M)$ can be computed using the recursive application of the following rule for each $l$:*

$$Z(M) = \sum_{k=0}^{n_l} \binom{n_l}{k} \left( Z(M_{l,k}^P) \right)$$

*Proof.* The proof follows immediately from the construction of the MLN $M_{l,k}^P$ and realizing that we apply the Binomial only over those constraint tuples taking values from the partition component $V_l^{x_j}$. From the partitioning property of $M$ each $V_l^{x_j}$ is disjoint from other partition components and hence, can be processed independently. Further, from the canonical form of constraints in $M$ and the count preserving property, each constant in the set $V_l^{x_j}$ is symmetrical to others and hence, we only need to keep an account of for how many of these, $P(x_j)$ groundings are true (or false). Hence, the proof follows. $\square$

### 4.1 Single Occurrence

Mittal et al. [3] proposed the single occurrence rule for lifting in MAP problems. An equivalence class of variables $\hat{x}$ is said to be single occurrence if any given formula $f_l \in F$ contains at most one occurrence of variables from the set $\hat{x}$. If $\hat{x}$ is a single occurrence variable in an MLN theory $M$, then the problem of finding the MAP solution of $M$ can be equivalently formulated as the problem of finding MAP over a reduced theory $M'$ such that a) For every formula $f_l \in F$ with weight $w_l$ in which a variable $x \in \hat{x}$ appears, there is a formula in $M'$ with weight $w_l$ b) For every formula $f_l \in F$ with weight $w_l$ in which a variable $x \in \hat{x}$ does not appear, $M'$ has a formula $f_l$ with weight $w_l/n$, $n$ being the domain size of variables in $\hat{x}$. c) Domain of every variable in the set $\hat{x}$ has been reduced to a single constant. The proof follows from the fact that there are $n$ equivalent reduced theories (up to renaming of a constant) whose groundings add up to groundings in $M$ and hence, the MAP solution for the original theory can be read off from any of the reduced theories $M'$. Note that since we are only care about the MAP solution, the reduced theory $M'$ can be equivalently constructed by having a weight of $w_l * n$ for the formulas containing a variable from the set $\hat{x}$ and the original weight $w_l$ for the remaining formulas.

Next, we will extend the above rule to work with constrained theories. We will perform a construction similar to the case of Binomial. Let $M$ be a constrained theory. Let $\hat{x}$ be single occurrence in $M$ and let $M$ be partitioned with respect to the set of variables in $\hat{x}$. Let $\mathcal{V}^{\hat{x}} = \{V_l^{\hat{x}}\}_{l=1}^r$ be the partition of $\hat{x}$ values in $M$. Let $A_l^x$ be a subset constraint with support given by the set $V_l^{\hat{x}}$. We will construct a reduced theory $M_l^{\hat{x}}$ from $M$ in the following manner. For every formula $f_l \in F$ be a formula in $M$ which contains variables from the set $\hat{x}$ a) Restrict $f_l$ to the constraint $\neg A_l^x$ (i.e. tuples not taking values for $\hat{x}$ variables from the set $V_l^{\hat{x}}$. b) For the remaining tuples, create a new formula $f_l'$ with weight $w_l'$ by:
1. Restricting the domain of $\hat{x}$ to a singleton for $A_l^x$
2. Assigning a weight of $w_l'{=}w_l * |V^{\hat{x}}|$ to the new formula
Intuitively, we have restricted the domain of partition component $V_l^{\hat{x}}$ to a single constant. We are now ready to define the single occurrence rule for constrained theories.

**Theorem.** *Let $M$ be a constrained MLN theory all of whose constraints are in canonical form. Let $\hat{x}$ denote an equivalence class of variables which is single occurrence in $M$. Let $\mathcal{V}^{\hat{x}} = \{V_l^{\hat{x}}\}_{l=1}^r$ denote the partition of $\hat{x}$ values in $M$. Then, the problem for finding MAP over $M$ can be solved by recursively reducing the theory to $M_l^{\hat{x}}$ for each $l$.*

*Proof.* The proof follows from the construction of the reduced theory $M_l\hat{x}$. $M_l\hat{x}$ applies the single occurrence decomposition only over those constraint tuples which take $\hat{x}$ values in the set $V_l^{\hat{x}}$. From the partitioning property of $M$ each $V_l^{\hat{x}}$ is disjoint from other partition components and hence, can be processed independently. Further, From the canonical form of constraints in $M$, we know that each constant in the set $V_l^{\hat{x}}$ is symmetric to each other and hence, single occurrence decomposition can be applied over this set of constants. This process is repeated in turn for every $V_l^{\hat{x}}$ to get the final decomposition. $\square$

**Theorem 4.3.** *Let $M$ be a constrained MLN theory. The application of the modified lifting rules over this constrained theory can be exponentially more efficient than first converting the theory in the normal form and then applying the original formulation of the lifting rules.*

*Proof.* We will prove the theorem by giving the template for constructing an example MLN where the result holds. Consider an MLN theory with a singleton predicate $P(x)$ with $Dom(x) = \{c_i\}_{i=1}^n$. Let $C_1 = \{c_i\}_{i=1}^{n/2}$ and $C_2 = \{c_i\}_{i=n/2+1}^n$ be a partition of the domain. Let there be full binary evidence on $P(x)$ with $P(c_i) = true, \forall c_i \in C_1$ and $P(c_i) = false, \forall c_i \in C_2$. And let this be the only evidence. Further, let there be no additional constraints. Let $\hat{x}$ be the equivalence class in which $x$ appears and let $\hat{x}$ be a decomposer. Then, partition of $\hat{x}$ in $M$ will have two components one each for the constraints of the form, $x \in C_1$ and $x \in C_2$ (all the values in each of the sets $C_1$ and $C_2$ are symmetric to each other). This will require two applications of the decomposer rule, one for each partition component, leading a complexity of $O(1)$. In contrast, to convert the theory in the normal form, we will have to first create ground predicates for every substitution of the constants appearing in the evidence i.e. $Dom(x)$. This will mean creating at least $n$ groundings for every predicate in which $\hat{x}$ appears, leading to a complexity of $O(2^n)$ (two possibilities for each of the $n$ groundings being $true$ or $false$). $\square$

## 4.2 Example

Now we will illustrate the application of different operations and rules presented in the paper using a simple example. Suppose we have a constrained MLN theory $M$ as follows :

$M = \{\{P(x_1) \vee Q(x_2), w_1, S^{\mathbf{x}}\}, \{P(y_1), w_2, S^{\mathbf{y}}\}\}$
Here $S^{\mathbf{x}} = \{T^{\mathbf{x}}\}, S^{\mathbf{y}} = \{T^{\mathbf{y}}\}$,
where $T^{\mathbf{x}} = \{x_1, x_2 \in \{1, 2, \ldots, 20\}, x_1 = x_2\}, T^{\mathbf{y}} = \{y_1 \in \{1, 2, \ldots, 15\}\}$

We wish to find the partition function $Z$ of the above MLN theory. For that, we do the following steps :

**1. Convert $M$ into canonical form :** $M$ is already in canonical form so we don't need to do anything.

**2. Apply binomial rule on $M$ :** Note that there is no decomposer in $M$, therefore we can't apply the decomposer rule at this stage. $P$ is a singleton predicate, so we can apply the binomial rule. The variables which appear as arguments of $P$ are $x_1$ and $y_1$. To apply the binomial rule, we need to partition the theory $M$ with respect to the variables $x_1$ and $y_1$. The original support sets of $x_1$ and $y_1$ are given by $\{1, 2, \ldots, 20\}$ and $\{1, 2, \ldots, 15\}$, respectively. After partitioning, we get the following support sets : $\{1, 2, \ldots, 15\}$ and $\{16, 17, \ldots, 20\}$. Consequently, the constraint tuples corresponding to the formulas split apart and the constraint sets can be described as follows:

$S^{\mathbf{x}} = \{T_1^{\mathbf{x}}, T_2^{\mathbf{x}}\}, S^{\mathbf{y}} = \{T^{\mathbf{y}}\}$
where $T_1^{\mathbf{x}} = \{x_1 \in \{1, 2, \ldots, 15\}, x_2 \in \{1, 2, \ldots, 20\}, x_1 = x_2\}$,
$T_2^{\mathbf{x}} = \{x_1 \in \{16, 17, \ldots, 20\}, x_2 \in \{1, 2, \ldots, 20\}, x_1 = x_2\}$, and
$T^{\mathbf{y}} = \{y_1 \in \{1, 2, \ldots, 15\}\}$

Note that after the application of the partition operation, the constraint tuples $T_1^{\mathbf{x}}$ and $T_2^{\mathbf{x}}$ are no longer in the canonical form since the variables $x_1$ and $x_2$, which participate in the equality constraint do not have identical support sets. Therefore we need to canonicalize the constraint tuples. After canonicalizing, the new constraint sets can be described as:

$S^{\mathbf{x}} = \{T_1^{\mathbf{x}}, T_2^{\mathbf{x}}\}, S^{\mathbf{y}} = \{T^{\mathbf{y}}\}$
where $T_1^{\mathbf{x}} = \{x_1 \, x_2 \in \{1, 2, \ldots, 15\}, x_1 = x_2\}$,
$T_2^{\mathbf{x}} = \{x_1, x_2 \in \{16, 17, \ldots, 20\}, x_1 = x_2\}$, and
$T^{\mathbf{y}} = \{y_1 \in \{1, 2, \ldots, 15\}\}$

Since the theory has been partitioned with respect to the variables $x_1$ and $y_1$, and is now in the canonical form, we can apply the binomial rule recursively on the two partition elements $\{1, 2, \ldots, 15\}$ and $\{16, 17, \ldots, 20\}$. Let $M_{l,k}^P$ be the constrained MLN obtained from $M$ by considering the $l^{th}$ partition element and setting $k$ groundings of the predicate $P$ to $true$ and remaining to $false$. Then

$Z(M) = \sum_{k=0}^{k=n} \binom{n}{k} Z(M_{l,k}^P)$ (refer Theorem 4.2)

In our example above, considering the $1^{st}$ partition element $\{1, 2, \ldots, 15\}$, during the application of the binomial rule, we get 16 constrained MLN theories $M_{1,0}^P, M_{1,1}^P, M_{1,2}^P, \ldots, M_{1,15}^P$. Therefore

$Z(M) = \sum_{k=0}^{k=15} \binom{15}{k} Z(M_{1,k}^P)$

For clarity, we describe one of the MLN theories $M_{1,5}^P$ obtained using steps as described in section 4.2. Recall that $M_{1,5}^P$ means 5 of the 15 groundings of Predicate $P$ are $true$ and the remaining 10 are $false$.

$M_{1,5}^P = \{\{P(x_1) \vee Q(x_2), w_1, S_1^{\mathbf{x}}\}, \{true \vee Q(x_2), w_1, S_2^{\mathbf{x}}\}, \{false \vee Q(x_2), w_1, S_3^{\mathbf{x}}\},$
$\{true, 5w_2, S_1^{\mathbf{y}}\}, \{false, 10w_2, S_2^{\mathbf{y}}\}\}$
Here $S_1^{\mathbf{x}} = \{T_{11}^{\mathbf{x}}\}, S_2^{\mathbf{x}} = \{T_{21}^{\mathbf{x}}\}, S_3^{\mathbf{x}} = \{T_{31}^{\mathbf{x}}\}, S_1^{\mathbf{y}} = S_2^{\mathbf{y}} = \{\}$
where $T_{11}^{\mathbf{x}} = \{x_1, x_2 \in \{16, 17, \ldots, 20\}, x_1 = x_2\}$,
$T_{21}^{\mathbf{x}} = \{x_2 \in \{1, 2, \ldots, 5\}\}, T_{31}^{\mathbf{x}} = \{x_2 \in \{6, 7, \ldots, 15\}\}$

First formula in the above theory represents the restriction of the theory $M$ to the partition element $\{16, 17, \ldots, 20\}$. Remaining formulas represent the restriction of $M$ to the partition element $\{1, 2, \ldots, 15\}$, where each of the original formula is split into two formulas corresponding to $P$ being $true$ or $false$. Note that since we are working on the partition element $\{1, 2, \ldots, 15\}$, this splitting of the formulas on $P$ being $true$ or $false$ happens only for the theory restricted to this

partition element. Weight of the formula $\{true, 5w_2, S_1^{\mathbf{y}}\}$ is $5w_2$, since the extension count of the constraint tuple $T^{\bar{y}_1}$ i.e., constraint tuple obtained after setting $P(y_1)$ to $true$ in the formula $\{P(y_1), w_2, S^{\mathbf{y}}\}$, is 5. Similarly the weight of the formula $\{false, 10w_2, S_1^{\mathbf{y}}\}$ is $10w_2$.

Next we recursively apply the binomial rule on the $2^{nd}$ partition $\{16, 17, \ldots, 20\}$ for each of the theories $M_{1,0}^P, M_{1,1}^P, M_{1,2}^P, \ldots, M_{1,15}^P$ obtained above. As before, let us consider the theory $M_{1,5}^P$. During the application of the binomial rule on the $2^{nd}$ partition $\{16, 17, \ldots, 20\}$, we get 6 constrained MLN theories $(M_{1,5}^P)_{2,0}, (M_{1,5}^P)_{2,1}, (M_{1,5}^P)_{2,2} \ldots (M_{1,5}^P)_{2,5}$. Below we show one of the MLN theories $(M_{1,5}^P)_{2,2}$. Let us call it $M'$. Note that in $M'$, we set 5 groundings of $P$ as $true$ for the partition $\{1, 2, \ldots, 15\}$, and 2 groundings of $P$ as $true$ for the partition $\{16, 17, \ldots, 20\}$. Therefore $M'$ can be written as :

$M' = \{\{true \vee Q(x_2), w_1, S_1^{\mathbf{x}}\}, \{false \vee Q(x_2), w_1, S_2^{\mathbf{x}}\}, \{true, 5w_2, S_1^{\mathbf{y}}\}, \{false, 10w_2, S_2^{\mathbf{y}}\}\}$
Here $S_1^{\mathbf{x}} = \{T_{11}^{\mathbf{x}} \vee T_{12}^{\mathbf{x}}\}, S_2^{\mathbf{x}} = \{T_{21}^{\mathbf{x}} \vee T_{22}^{\mathbf{x}}\}, S_1^{\mathbf{y}} = S_2^{\mathbf{y}} = \{\}$,
where $T_{11}^{\mathbf{x}} = \{x_2 \in \{1, 2, \ldots, 5\}\}, T_{12}^{\mathbf{x}} = \{x_2 \in \{16, 17\}\}$,
$T_{21}^{\mathbf{x}} = \{x_2 \in \{6, 7, \ldots, 15\}\}, T_{22}^{\mathbf{x}} = \{x_2 \in \{18, 19, 20\}\}$

Note that we have merged those constrained formulas together which have the same underlying first order formula but possibly different constraint sets.

**3. Apply Decomposer :** Note that we can split the theory $M'$ into two independent theories $M''$ and $M'''$ given as follows :

$M'' = \{\{true \vee Q(x_2), w_1, S_1^{\mathbf{x}}\}, \{False \vee Q(x_2), w_1, S_2^{\mathbf{x}}\}\}$
Here $S_1^{\mathbf{x}} = \{T_{11}^{\mathbf{x}} \vee T_{12}^{\mathbf{x}}\}, S_2^{\mathbf{x}} = \{T_{21}^{\mathbf{x}} \vee T_{21}^{\mathbf{x}}\}$
where $T_{11}^{\mathbf{x}} = \{x_2 \in \{1, 2, \ldots, 5\}\}, T_{12}^{\mathbf{x}} = \{x_2 \in \{16, 17\}\}$,
$T_{21}^{\mathbf{x}} = \{x_2 \in \{6, 7, \ldots, 15\}\}, T_{22}^{\mathbf{x}} = \{x_2 \in \{18, 19, 20\}\}$

$M''' = \{\{true, 5w_2, S_1^{\mathbf{y}}\}, \{false, 10w_2, S_2^{\mathbf{y}}\}\}$
Here $S_1^{\mathbf{y}} = S_2^{\mathbf{y}} = \{\}$

Since $M''$ and $M'''$ do not share any ground atom, they can be dealt with independently. Let us consider $M''$ first. We can apply the decomposer rule on the equivalence class $\{x_2\}$. Note that this theory is already partitioned with respect to $\{x_2\}$ since the corresponding support sets (partition elements) $\{1, 2, \ldots, 5\}, \{6, 7, \ldots, 15\}, \{16, 17\}$, and $\{18, 19, 20\}$ are disjoint with each other. We will use $V_l$ to denote the $l^{th}$ partition element where $1 \leq l \leq 4$. Let $M_l''$ denote the theory obtained by restricting $M''$ to a singleton $\{v\}$, where $v \in V_l$. Then

$Z(M'') = \Pi_{l=1}^4 (Z(M_l''))^{|V_l|}$ (refer theorem 4.1).

For illustration, $M_2''$ can be described as :

$M_2'' = \{\{false \vee Q(x_2), w_1, S_1^{\mathbf{x}}\}\}$
Here $S_1^{\mathbf{x}} = \{T_{11}^{\mathbf{x}}, \text{where } T_{11}^{\mathbf{x}} = \{x_2 \in \{6\}\}$

Note that the support set of $x_2$ has been reduced to a single constant due to the application of the decomposer rule. We can perform a similar reduction for the other partition elements. Therefore

$Z(M'') = (Z(M_1''))^5 * (Z(M_2''))^{10} * (Z(M_3''))^2 * (Z(M_4''))^3$

Note that each of the theories on R.H.S. above is now propositional and thus can be solved easily using a propositional solver.

Next let us consider the theory $M'''$. The first formula in $M'''$ evaluates to $true$, and the second formula evaluates to $false$. Therefore applying the base case, we get

$Z(M''') = e^{5w2} + 1$

Finally, since $M''$ and $M'''$ are independent of each other :

$Z(M') = Z(M'') * Z(M''')$

| Domain | Source | Rules | Type (# of const.) | Evidence |
|---|---|---|---|---|
| Professor & Student (PS) | [1] | GoodStud($s$) $\wedge$ GoodProf($p$) $\wedge$ AdvBy($s,p$) $\Rightarrow$ FutrProf($s$); AdvBy($s,p$)$\Rightarrow$ CoAuth($s,p$) | prof (100) student (**var**) | GoodStud GoodProf |

Table 2: Dataset Details. **var** means the domain of this type was varied in experiments. Non-evidence predicates were modeled as Query.

## 5   Experiments

**Professor and Student (PS) :**  The details of dataset are provided in table 2. Figure 1a shows the results as the domain size of $student$ is varied from 100 to 800 with 40% evidence on $student$ in the PS domain. Evidence on $prof$ is 50% in each case. The time taken by Normal grows exponentially ($y$-axis is on log scale) whereas SetInEq scales almost linearly. GCFOVE fails to run even on domain size of 100 of $student$. Figure 1b plots the number of nodes expanded by Normal and SetInEQ. In SetInEq, no. of nodes almost remain constant whereas in Normal, no. of nodes increase linearly. Figure 1c shows the time taken by SetInEq and Normal algorithms as we vary the evidence on $student$ with a fixed domain size of 300. Evidence on $prof$ was not varied, keeping it fixed at 50%. The time required by SetInEq stays almost the constant whereas Normal grows exponentially ($y$-axis is on log scale).

(a) Domain size vs time (seconds). Y-axis is on log scale

(b) Domain size vs no. of nodes expanded.

(c) Evidence percentage vs time (seconds). Y-axis is on log scale

Figure 1: PS domain

## Footnotes

[1] $v_i$ satisfies all the required constraints that $x_i$ is involved in, by virtue of being equal to $v_j$