[Reviews · NeurIPS 2015]

Submitted by Assigned_Reviewer_1

The paper considers lifted inference within probabilistic relational models. It proposes a novel constraint language that allows one to use subset, equality, and inequality constraints for encoding substitutions over the variables in the theory.

For this language, the paper shows how to reformulate lifted inference methods based on sampling and search and shows

benefits over existing methods.

The idea presented in the paper is really nice.

I have enjoyed reading it. It is quite neat to see that the lifted inference community should start working on more expressing constraint languages to represent the "equivalence classes". I guess this could also be

rephrased as, the amount of symmetry exploitable depends on the language you consider to exploit symmetries. Really nice.

Consequently, I have only some minor comments.

First, it would be great if the authors could explain in more detail the statement "Projection operation has a linear time

complexity once the extension count property has been ensured

using splitting as described above." How many splits to we

have to make? What is the overall worst-case complexity? Isn't it that we may still have to ground everything? If so, this should be said to avoid to give the impression we have solved lifted inference even beyond the barriers proven in the literature. This overall complexity should also be presented for the assumptions/preprocessing steps presented on page 5 (upper parts).

Second, given the space left for references, the authors should provide a more updated list of references on lifted

probabilistic inference.

Martin Mladenov, Kristian Kersting:

Equitable Partitions of Concave Free Energies.

UAI 2015

Martin Mladenov, Amir Globerson, Kristian Kersting: Lifted Message Passing as Reparametrization of Graphical Models

UAI 2014

Udi Apsel, Kristian Kersting, Martin Mladenov: Lifting Relational MAP-LPs Using Cluster Signatures.

AAAI 2014

all can work with subset (hypercube), no inequlity, intersection,

union.

Summary: + speepding up probabilistic inference is an important research direction + the paper shows that working on the constraint language used to encode diagrams for inference is important and can lead to much better scaling

Submitted by Assigned_Reviewer_2

In the experimental setup (Table 2), it is misleading to have only evidence on unary relations for the first two domains. It's show in [8] that unary evidence can always be handled efficiently and that the KC constraint language can be used to do inference in, for example, friends and smokers with unary evidence, in the same time regardless of the amount of evidence and polynomial in the domain size (see [8, Figure 5]). Therefore Section 5.1 is not very meaningful; that setting is entirely solved. This makes me wonder how the proposed constraint language performs with binary evidence which is much harder to deal with (and probably breaks all symmetries).

The ProbLog 2 system used as a baseline has nothing to do with lifted inference! It contains no lifted inference code and has nothing to do with the following reference [9]! There is an implementation of the algorithms in [9], but that is an entirely different system called WFOMC.

In table 1, it's unclear what is meant by tractable solver. For KC and PTP, why are the constraint solvers intractable? They are simply conjunctions of simple set operations that can be evaluated efficiently, as far as I understand?

In table 1, being able to write union does not really affect the expressivity of the constraint language: we can always take a constrained formula (FORALL C union D, Formula) with a union and write it without a union by using conjunction at the outermost level: (FORALL C, Formula) AND (FORALL D, Formula).
Summary: Constraint languages in lifted inference deserve more study, and this paper makes good contributions towards a more general representation. The comparison with related work is fair. Only the experiments have some mistakes and are maybe not that meaningful.

Submitted by Assigned_Reviewer_3

Summary:

Lifted inference enables tractable inference in large probabilistic models by exploiting symmetries and independencies over variables. This paper provides a comprehensive method to (a) specify constraints; (b) use them for lifted inference; and (c) produce constraints from evidence. The authors first introduce setineq a constraint language for specifying symmetries and independencies using set membership and (in)equality constraints, and supply basic operators for manipulating these constraints. Next, The authors develop algorithms to use these constraints and basic operators to produce lifting rules (Decomposer/Binomial). Finally, the authors present a greedy method for deriving constraints from evidence. The authors test a Java-based implementation of the setineq-based constraint specification against other lifted inference implementations and show superior scaling and equivalent output to unconstrained representations.

Quality:

This paper has a number of technical contributions in representation and implementation of lifted inference methods. Experimental evaluation uses standard benchmark datasets, seems to provide fair comparisons to related work, and shows impressive performance.

Clarity:

My main issues with this paper relate to clarity. The paper relies heavily on the supplemental material for proofs, algorithms, and the Single Occurrence rule, and including some of this supplemental material in the main paper may be helpful.

Since the paper covers a great deal of ground, each contribution must be framed carefully. The formal specification of setineq and the join/partition operators are clearly presented and benefits from many examples to help the reader understand how constraints are specified. However, the remainder of the paper would be improved with restructuring, proofreading (see below), and additional examples. One potential improvement would be moving Section 4.4 to section 3, since is relatively independent of the lifted inference rules and strongly relates to producing a constraint tuple from evidence. Additionally, revising section 4.4 may be helpful - the section is very terse and it's difficult to distinguish the paper's contributions from the previous work. In the lifted inference rules, the description of the Binomial algorithm (4.3) is extremely difficult to read. Examples for Theorem 4.1 and 4.2 would lend the preceding (somewhat dense) exposition some intuitive appeal.

Originality:

To the best of my knowledge, this work is original and relevant related work is cited.

Significance:

My understanding is that this provides an incremental refinement to lifted inference, and provides a starting point for deeper investigation. The empirical results seem very strong, and I wouldn't be surprised to see following work building more advanced lifted inference methods on setineq.

Typos:

Bell number (210)

grammar (240)

set definition (253)

set V^y_l (254)

argumene (297)

nmw (304)

Professors and Students (365)

Summary: Probabilistic models increasingly employ lifted inference to scale to large problem domains and this paper introduces a constraint language and translation methods to produce compact, lifted representations of problems. The paper is (mostly) well written, provides meaningful technical contributions and has thorough empirical analysis.

Submitted by Assigned_Reviewer_4

Recently there has been much interest in the relational learning community in lifted inference, leading to a greater focus on the particular language used to define first order models, because different languages enable different lifted inference techniques to different degrees. This paper proposes a new language, SetInEQ. SetInEQ can be thought of as a generalization of Markov logic in that it provides mechanisms for restricting the groundings of rules to specific sets of constants. While in principle such constrained MLNs could be represented in an unconstrained but more complex normal form, explicitly representing and levering constraints can enable faster lifted inference.

SetInEQ allows for restricting grounds to constants that are either in or not in a given set. This generalizes several existing constraint languages. The paper then shows how several common lifted inference techniques can be applied to a SetInEQ model. The experimental results appear to show that inference with SetInEQ is faster than inference with other constraint languages, ProbLog, and inference with MLNs converted to normal form using several software packages.

Questions:

The paper is unclear to me on several important points, and so I cannot recommend it more strongly for publication until they are cleared up.

First, can the authors elaborate on what makes SetInEQ more efficient than other constraint languages? Based on Table 1, it seems that existing languages like GFCOVE and approximate LBP support everything that SetInEQ does except inequality constraints. How does that difference lead to the performance differences in the experiments?

Second, can the authors elaborate on how the test problems are represented in the constraint languages? I think it can be inferred from the Evidence Processing paragraph, but is it correct to infer that evidence is the only type of constraint in the experiments? If so, that could be clarified in the writing. Also, then why do the results differ when the evidence % is zero? What is the difference between the methods in Section 5.2?

Which lifted inference rule or rules are most important for achieving the results? Based on the structure of the test problems, how does each rule in isolation compare against its implementation in other constraint languages and normal form? This does not seem to be evaluated.
Summary: The paper proposes a new constraint language, SetInEQ, for constraining the groundings of rules in Markov logic networks (MLNs). The emphasis of the language's design is to support tractable lifted inference with both search and sampling algorithms. On three common data sets, lifted inference with SetInEQ is faster than inference with other constraint languages, as well as generic lifted inference algorithms applied to MLNs converted to normal form. The paper is good, but it is unclear on several points, which prevents me from recommending it more highly.

Author Feedback
Author rebuttal: We thank all the reviewers for helpful comments.

Reviewer 1:

Let k be the number of variables involved in inequality constraints. Number of splits is O(bell_number_k)) (see [14]). When dealing with models such as MLNs, k is at most the number of variables in a formula, which is typically very small (single digit). For each split, the complexity of projection is linear in the size of the constraint, therefore the total complexity is O(l*bell_number(k)), l being the size of the constraint. This operation does not require explicit grounding of the formula over which constraint is specified.

The complexity of converting into canonical form is already presented in section 3. The complexity of other two assumptions is linear in the size of the theory.

Let y denote the set of variables being partitioned or being restricted. Let cnt denote the total number of constraints in which y variables appear. Let n be the number of constants appearing in those constraints. Partitioning can be implemented in O(n*cnt) time assuming constant time hashing. The complexity of the restriction operation is O(|y|*cnt*|V|) where |V| is the size of the support set for the restriction.

We will add the complexity results as well as missing references to the camera ready.

Reviewer 2:

Thanks for the pointer to the result in [8]. We believe same is the case with our constraint language (though we do not formally prove it). Intuitively, the unary evidence can be partitioned into symmetrical groups corresponding to true/false/unknown groundings. Each of these cases can be handled separately and our algorithm can take advantage of this (see Motivation in Section 4.1). Figure 1(b) illustrates this. There is some variation observed across varying evidence due to pre-processing of the unary evidence. Increasing evidence simplifies the theory and computational time goes down further. Our results are meaningful since the stated result was proven only for knowledge compilation based approaches to the best of our knowledge.

Out of the three datasets presented in Table 2, only FS involves unary evidence. The other two datasets i.e., WebKB and IMDB(Section 5.2) are over binary evidence. Our results over these two datasets clearly demonstrate that symmetries can help to a great extent even when dealing with non-unary evidence.

Thanks for the note regarding Problog 2. We will fix it in the final version.

The intractability is hidden in handling the inequality constraints. Both these approaches make a call to a general purpose constraint solver, which can be quite expensive.

Not being able to write union requires one to replicate the same formula multiple times for each of the constraints in the union. This makes the identification of which lifting rules to apply significantly more complex.

Reviewer 3:

GCFOVE can represent subset constraints like ours though there is no setout constraint (see Section 3). GCFOVE is based on lifted VE whereas our approach applies constraints to lifted inference rules. These can be significantly faster than lifted VE based approaches explaining the difference in performance. This is also shown by Gogate and Domingos (2011); but they do not handle any subset constraints. We do not compare with LBP. It is an approximate inference algorithm and does not have any convergence guarantees.

Yes, in the current experiments, evidence is the only type of constraint. We will make it explicit in the final version. In Section 5.2, the two algorithms compared are Normal and SetInEq. Both of them perform identically at 0% evidence (see Figure 2(b)). The difference in performance at non-zero evidence stems from the fact our approach can better exploit the evidence symmetry (Normal splits apart the evidence completely). In Section 5.1 too, SetInEq and Normal are identical at 0% evidence (Figure 1(b)). GCFOVE and SetInEq (Figure 1(b)) perform differently because they are different lifted inference algorithms. GCFOVE fails to scale on PS domain even at 0% evidence due to the same reason (see supplement).

Our algorithm could take advantage of both the decomposer and the binomial rules over the existing approaches for each dataset.Decomposer rule led to larger exploitation of symmetries since (sub)domain size was reduced to one. In all the cases, decomposer was applicable only after one or more applications of the binomial. In our additional experiments, though both rules were found useful when applied in isolation, their combination was significantly more effective. We can add details in the final version.

Reviewer 4:

We will re-structure the paper to improve clarity. We will also add more details (and add an example, space permitting) and clarify our contributions (Section 4.4) for better understanding. We will also fix minor typos.